**Subject Area:**
biochemistry/cellular biology

kinetochore, kinetoplastid, *Trypanosoma brucei*, centromere, Gcn5-related *N*-acetyltransferase

**Author for correspondence:**
Bungo Akiyoshi
e-mail: bungo.akiyoshi@bioch.ox.ac.uk

# Identification of four unconventional kinetoplastid kinetochore proteins KKT22–25 in *Trypanosoma brucei*

Olga O. Nerusheva, Patryk Ludzia and Bungo Akiyoshi

Department of Biochemistry, University of Oxford, Oxford OX1 3QU, UK

BA, 0000-0001-6010-394X

The kinetochore is a multi-protein complex that drives chromosome segregation in eukaryotes. It assembles onto centromere DNA and interacts with spindle microtubules during mitosis and meiosis. Although most eukaryotes have canonical kinetochore proteins, kinetochores of evolutionarily divergent kinetoplastid species consist of at least 20 unconventional kinetochore proteins (KKT1–20). In addition, 12 proteins (KKT-interacting proteins 1–12, KKIP1–12) are known to localize at kinetochore regions during mitosis. It remains unclear whether KKIP proteins interact with KKT proteins. Here, we report the identification of four additional kinetochore proteins, KKT22–25, in *Trypanosoma brucei*. KKT22 and KKT23 constitutively localize at kinetochores, while KKT24 and KKT25 localize from S phase to anaphase. KKT23 has a Gcn5-related *N*-acetyltransferase domain, which is not found in any kinetochore protein known to date. We also show that KKIP1 co-purifies with KKT proteins, but not with KKIP proteins. Finally, our affinity purification of KKIP2/3/4/6 identifies a number of proteins as their potential interaction partners, many of which are implicated in RNA binding or processing. These findings further support the idea that kinetoplastid kinetochores are unconventional.

## 1. Introduction

Kinetoplastids are a group of unicellular flagellated eukaryotes found in diverse environmental conditions [1]. It has been proposed that kinetoplastids may represent one of the earliest-branching eukaryotes based on a number of unique molecular features [2]. Understanding their biology could therefore provide insights into the extent of conservation or divergence among eukaryotes and lead to a deeper understanding of biological systems. Importantly, three neglected tropical diseases are caused by parasitic kinetoplastids: African trypanosomiasis, Chagas disease and leishmaniasis, caused by *Trypanosoma brucei*, *Trypanosoma cruzi* and *Leishmania* spp., respectively [3]. Although recent advances in public health and combination therapy have decreased the effect of these diseases, new drugs and druggable pathways are still very much needed against these diseases [4]. To this end, a more thorough understanding of the unique underlying biological mechanisms of kinetoplastids is critical.

One such fundamental process is the transmission of genetic material from mother to daughter cells, which is crucial for the survival of all organisms. Chromosome segregation in eukaryotes is driven by the kinetochore, a macromolecular protein complex that assembles onto centromeric DNA and captures spindle microtubules during mitosis [5]. Its structural core is typically composed of DNA-binding and microtubule-binding modules [6]. At least a fraction of core kinetochore proteins are present in nearly all sequenced eukaryotes, implying that most eukaryotes use a largely conserved mechanism of DNA and microtubule binding [7–9]. However, none of canonical kinetochore proteins have been identified in the genome of kinetoplastids [10,11]. To identify their kinetochore components, we previously carried out a YFP-tagging screen and identified a

protein that forms kinetochore-like dots [12]. Affinity purification of this protein identified co-purifying proteins whose localizations were subsequently examined by microscopy. This process was repeated until saturation, leading to the identification of 20 proteins that localize at kinetochores in *T. brucei*. Chromatin immunoprecipitation followed by deep sequencing confirmed that they are indeed kinetochore proteins, and we therefore named them KKT1–20 (kinetoplastid kinetochore proteins). Although these proteins are highly conserved among kinetoplastids, their apparent orthologues are absent in other eukaryotes, suggesting that kinetoplastids have an unconventional type of kinetochore proteins [12,13].

Recently, KKT-interacting protein 1 (KKIP1) was identified as a protein distantly related to Ndc80/Nuf2 microtubule-binding proteins based on weak similarity in the coiled-coil regions [14]. However, KKIP1 apparently lacks the calponin homology (CH) domain, a critical feature of Ndc80/Nuf2 proteins. KKIP1 therefore does not appear to be a genuine Ndc80/Nuf2 orthologue. Nonetheless, KKIP1 localizes at kinetochores and its depletion causes severe chromosome segregation defects [14]. Immunoprecipitation of KKIP1 from chemically cross-linked cells led to the identification of six additional proteins (KKIP2–7) that localize to kinetochore regions during mitosis [14]. Very recently, immunoprecipitation of KKIP2–7 from non-cross-linked cells identified a nine-subunit protein complex called the KOK (kinetoplastid outer kinetochore) complex that consists of KKIP2, 3, 4, 6, 8, 9, 10, 11 and 12 [15]. KKT proteins were not detected in the immunoprecipitates of KKIP2–12 or KKIP1 without chemical cross-linking [15]. It therefore remains unclear whether KKIP1–12 interact with KKT proteins in native conditions.

A hallmark of kinetochores in most eukaryotes is the presence of specialized nucleosomes containing the centromere-specific histone H3 variant CENP-A, which epigenetically specifies the position of kinetochore assembly, forms the primary anchorage point to DNA and recruits other kinetochore proteins [16]. However, CENP-A is absent in kinetoplastids. It therefore remains unknown how their kinetochores assemble specifically at centromeres. *Trypanosoma brucei* has 11 large chromosomes that have regional centromeres of 20–120 kb in size, as well as approximately 100 small chromosomes that lack centromeres [17–19]. Although kinetochore assembly sites on large chromosomes are apparently determined in a sequence-independent manner, the underlying mechanism remains a mystery.

To understand how unconventional kinetoplastid kinetochores perform conserved functions such as kinetochore specification, it is critical to have a complete constituent list. In this study, we report the identification of four additional kinetochore proteins in *T. brucei*.

# 2. Results

## 2.1. Identification of KKT22 and KKT23 in *Trypanosoma brucei*

Our previous immunoprecipitation of KKT3 that was N-terminally tagged with YFP (YFP-KKT3) did not result in co-purification of other kinetochore proteins [12]. To verify this result, we made a strain that had a C-terminally YFP-tagged KKT3 (KKT3-YFP) as the sole copy of KKT3 and performed its immunoprecipitation using the same protocol. We detected a number of kinetochore proteins by mass spectrometry (figure 1*a*; electronic supplementary material, table S1), suggesting that YFP-KKT3 was not fully functional. In addition to known kinetochore proteins, there were two uncharacterized proteins (ORF Tb927.9.6420 and Tb927.10.6600) that co-purified with KKT3-YFP in an apparently specific manner (figure 1*a*). We tagged these proteins with an N-terminal YFP at the endogenous locus and found that they localized at kinetochores throughout the cell cycle (figure 1*b*,*c*). Immunoprecipitation of these proteins showed that they specifically co-purified with other KKT proteins (figure 1*d*,*e*). We therefore named them KKT22 and KKT23. These proteins were not detected in the immunoprecipitates of any other kinetochore protein [12], so it is likely that these proteins are closely associated with KKT3.

Homology search of KKT22 identified apparent orthologues in several kinetoplastids (table 1), but not in *Bodo saltans* (a free-living kinetoplastid) or other eukaryotes. A profile–profile comparison using HHpred [20] did not reveal any obvious domain, except for a possible zinc hook motif of Rad50 (electronic supplementary material, figure S1). KKT23 has a Gcn5-related *N*-acetyltransferase (GNAT) domain [21–23], which is not found in any known kinetochore protein in other eukaryotes. In humans, TIP60 and KAT7/HBO1/MYST2 acetyltransferases (both are members of the MYST subfamily) are known to regulate kinetochore functions but are not part of core kinetochores [24,25]. We found that KKT23 co-purified with many KKT proteins (figure 1*e*), implying that it is a core kinetochore protein in *T. brucei*. Interestingly, our sequence analysis failed to identify an obvious orthologous relationship with known GNAT subfamily members, suggesting that KKT23 forms a distinct subfamily. Our finding that an apparent orthologue of KKT23 is found even in divergent kinetoplastids (*B. saltans* and *Perkinsela*; table 1; electronic supplementary material, figure S2) raises a possibility that it plays a fundamental role at the kinetoplastid kinetochore, which warrants further investigation.

## 2.2. Identification of KKT24

In our purification of YFP-KKT22, there was another kinetochore protein candidate (ORF Tb927.10.4200; figure 1*d*). We found that this protein, in fact, localized at kinetochores from S phase to anaphase (figure 2*a*) and its immunoprecipitation confirmed specific co-purification with other kinetochore proteins (figure 2*b*). We therefore named it KKT24. Interestingly, KKT24 and KKIP1 share several similarities. Both proteins are predicted to consist mostly of coiled coils (electronic supplementary material, figure S3) [14], and their N-termini are located at the outer region of kinetochores, as judged by the formation of pairs of dots in metaphase (figure 2*a*) [15,26]. However, our immunoprecipitation data do not support a possibility that KKT24 and KKIP1 form a stable complex (figure 2*b* and see below). We also note that obvious orthologues for KKT24 and KKIP1 are not found in free-living *B. saltans*, an organism that has essentially all of KKT1–20 proteins (table 1) [12,13].

## 2.3. Identification of KKT25

Our purification of KKT24 led to the identification of another kinetochore protein candidate (ORF Tb927.8.2830) (figure 2*b*), which indeed localized at kinetochores from S phase to anaphase (figure 3*a*). We confirmed that this protein co-purified

royalsocietypublishing.org/journal/rsob Open Biol. 9: 190236

(a)

| proteins co-purified with KKT3-YFP | Mascot score | no. peptides | coverage (%) |
|---|---|---|---|
| KKT3 | 4112 | 194 | 49.6 |
| KKT2 | 1096 | 39 | 35.9 |
| KKT7 | 961 | 43 | 52.0 |
| KKT1 | 788 | 32 | 19.9 |
| KKT8 | 395 | 12 | 37.3 |
| KKT9 | 208 | 8 | 24.3 |
| KKT10 | 188 | 7 | 18.9 |
| KKT6 | 175 | 13 | 33.7 |
| KKT4 | 133 | 6 | 21.2 |
| KKT19 | 132 | 6 | 20.5 |
| KKT14 | 127 | 4 | 14.2 |
| Tb927.9.6420 (KKT22) | 119 | 6 | 33.9 |
| Tb927.10.6600 (KKT23) | 105 | 7 | 33.3 |
| KKT11 | 101 | 4 | 21.9 |

(b)

| | YFP-KKT22 | tdTomato-KKT2 | DAPI |
|---|---|---|---|
| G1 (1K1N) | | | K    N |
| S (1K*1N) | | | |
| G2/ prometaphase (2K1N) | | | |
| metaphase | | | |
| anaphase (2K2N) | | | |

(c)

| | YFP-KKT23 | tdTomato-KKT2 | DAPI |
|---|---|---|---|
| G1 (1K1N) | | | |
| S (1K*1N) | | | |
| G2/ prometaphase (2K1N) | | | |
| metaphase | | | |
| anaphase (2K2N) | | | |

(d)

| proteins co-purified with YFP-KKT22 | Mascot score | no. peptides | coverage (%) |
|---|---|---|---|
| KKT3 | 9597 | 426 | 38.2 |
| KKT7 | 7351 | 273 | 49.1 |
| KKT1 | 4377 | 124 | 15.0 |
| KKT8 | 3940 | 103 | 34.8 |
| KKT2 | 3553 | 104 | 22.1 |
| KKT4 | 1595 | 40 | 19.8 |
| KKT6 | 1489 | 42 | 25.9 |
| KKT23 | 1045 | 38 | 7.8 |
| KKT10 | 999 | 46 | 16.8 |
| KKT14 | 815 | 39 | 16.2 |
| KKT5 | 640 | 23 | 15.4 |
| KKT9 | 579 | 35 | 32.2 |
| KKT19 | 533 | 26 | 22.1 |
| KKT12 | 394 | 11 | 15.0 |
| Tb927.10.4200 (KKT24) | 178 | 11 | 6.0 |
| KKT22 | 85 | 10 | 15.8 |

(e)

| proteins co-purified with YFP-KKT23 | Mascot score | no. peptides | coverage (%) |
|---|---|---|---|
| KKT3 | 5560 | 258 | 39.3 |
| KKT7 | 1775 | 62 | 37.9 |
| KKT22 | 1480 | 67 | 45.1 |
| KKT1 | 897 | 30 | 16.3 |
| KKT8 | 864 | 24 | 41.9 |
| KKT2 | 756 | 21 | 15.6 |
| KKT4 | 708 | 13 | 15.8 |
| KKT6 | 386 | 9 | 30.7 |
| KKT23 | 345 | 12 | 37.4 |
| KKT9 | 329 | 16 | 31.3 |
| KKT19 | 295 | 8 | 32.6 |
| KKT14 | 207 | 6 | 20.7 |
| KKT10 | 201 | 7 | 15.7 |
| KKT5 | 201 | 5 | 21.1 |
| KKT11 | 101 | 3 | 12.0 |
| KKT12 | 95 | 2 | 14.5 |

**Figure 1.** Identification of KKT22 and KKT23. (a) KKT3-YFP co-purifies with a number of KKT proteins, including two kinetochore protein candidates, KKT22 and KKT23. See electronic supplementary material, table S1 for all proteins identified by mass spectrometry. Cell line, BAP1123. (b) YFP-KKT22 and (c) YFP-KKT23 localize at kinetochores throughout the cell cycle. K and N represent the kinetoplast (mitochondrial DNA) and nucleus, respectively. These organelles have distinct replication and segregation timings and serve as good cell-cycle markers. K* is an elongated kinetoplast and indicates that the nucleus is in S phase. Fluorescent protein signals were directly detected by microscopy. Similar results were obtained with KKT22-YFP and KKT23-YFP (data not shown). Bars, 5 μm. BAP1454 and BAP1593. (d) KKT22 co-purifies with KKT proteins and another kinetochore protein candidate, KKT24. BAP1490. (e) KKT23 co-purifies with KKT proteins. BAP1549.

with various kinetochore proteins (figure 3b) and therefore named it KKT25. Like KKT22 and KKT24, it is conserved in many kinetoplastids, but not in B. saltans or other eukaryotes (table 1; electronic supplementary material, figure S4). We failed to identify any obvious domain or predicted coiled coils in KKT25.

## 2.4. KKIP1 co-purifies with KKT proteins, not with KKIP proteins

A previous study by D'Archivio & Wickstead [14] identified a putative kinetochore protein KKIP1 that localized to kineto-chores. Its immunoprecipitation from chemically cross-linked

royalsocietypublishing.org/journal/rsob    *Open Biol.* **9**: 190236

**Table 1.** Conservation of KKT22–25, KKIP1 and KKIP5 among kinetoplastids.

| | KKT22 | KKT23 | KKT24 | KKT25 | KKIP1 | KKIP5 |
|---|---|---|---|---|---|---|
| *T. brucei* | Tb927.9.6420 | Tb927.10.6600 | Tb927.10.4200 | Tb927.8.2830 | Tb927.5.4520 | Tb927.7.6630 |
| *T. congolense* | | TcIL3000_10_5670 | TcIL3000_10_3510 | TcIL3000_0_41350 | TcIL3000_5_5210 | TcIL3000_0_33690 |
| *T. grayi* | DQ04_05731010 | DQ04_07341010 | DQ04_12781000 | DQ04_02041070 | DQ04_01751000 | DQ04_10231010 |
| *T. vivax* | | TvY486_1006570 | TvY486_0019400 | TvY486_0802340 | TvY486_0503940 | TvY486_0706450 |
| *T. cruzi* | TcCLB.506509.60 | TcCLB.510187.340 | TcCLB.511467.50 | TcCLB.504427.170 | TcCLB.509539.40 | TcCLB.510055.70 |
| *T. rangeli* | TRSC58_06150 | | TraAM80_06991 | TRSC58_01409 | TRSC58_03083 | TRSC58_00449 |
| *T. theileri* | TM35_000182020 | TM35_000022840 | TM35_000421720 | TM35_000132550 | TM35_000061590 | TM35_000202190 |
| *Blechomonas* | Baya_113_0090 | Baya_001_0150 | Baya_039_0490 | Baya_024_0050 | Baya_086_0290 | Baya_072_0080 |
| *Crithidia* | | CFAC1_250031900 | CFAC1_240027400 | CFAC1_150014500 | CFAC1_020006200 | CFAC1_090010600 |
| *Leptomonas* | | Lsey_0046_0080 | Lsey_0122_0130 | Lsey_0053_0040 | Lsey_0120_0220 | Lsey_0154_0070 |
| *Endotrypanum* | | EMOLV88_360026200 | EMOLV88_350006300 | EMOLV88_230010500 | EMOLV88_050005000 | EMOLV88_170008400 |
| *L. mexicana* | LmxM.15.0825 | LmxM.36.2100 | LmxM.34.0180 | LmxM.23.1610 | LmxM.05.0010 | LmxM.17.0430 |
| *Phytomonas* | GSEM1_T00007051001 | GSEM1_T00000676001 | GSEM1_T00004833001 | GSEM1_T00006110001 | GSEM1_T00003924001 | GSEM1_T00002251001 |
| *Paratrypanosoma* | PCON_0021420 | PCON_0016010 | PCON_0019770 | PCON_0076260 | PCON_0068010 | |
| *B. saltans* | | BSAL_82255 | | | | |
| *Perkinsela* | | XU18_2502 | | | | |

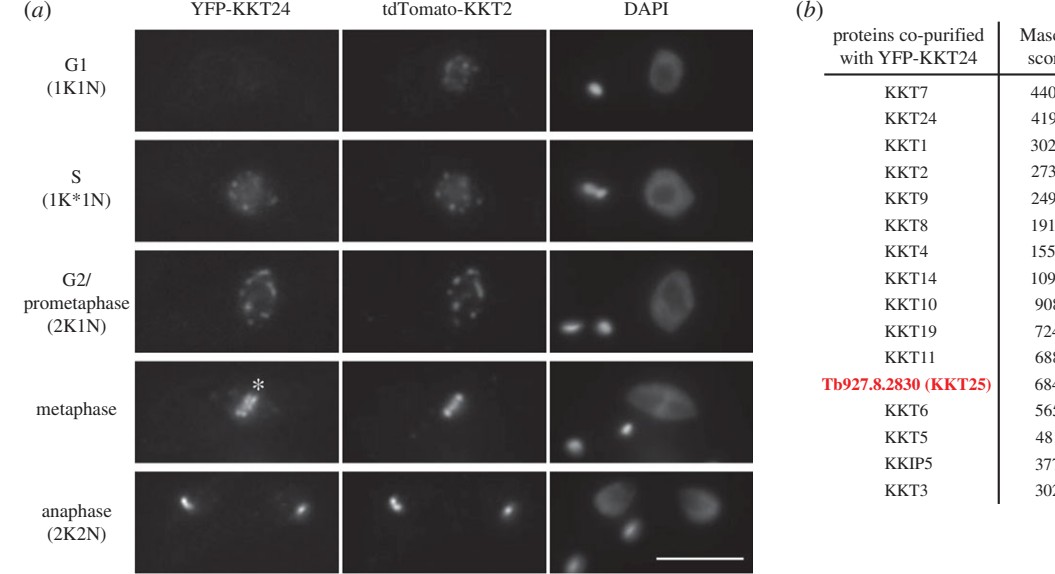

**Figure 2.** Identification of KKT24. (*a*) KKT24 localizes at kinetochores from S phase to anaphase. Note that YFP-KKT24 signals appear as pairs of dots (indicated by asterisk). Fluorescent protein signals were directly detected by microscopy. Bar, 5 μm. Cell line, BAP1819. (*b*) KKT24 co-purifies with a number of KKT proteins, KKIP5, and another kinetochore protein candidate, KKT25. BAP1635.

Figure 2(b) table:

| proteins co-purified with YFP-KKT24 | Mascot score | no. peptides | coverage (%) |
|---|---|---|---|
| KKT7 | 4400 | 153 | 69.3 |
| KKT24 | 4191 | 126 | 60.2 |
| KKT1 | 3024 | 106 | 27.5 |
| KKT2 | 2735 | 88 | 38.7 |
| KKT9 | 2495 | 88 | 78.4 |
| KKT8 | 1913 | 52 | 73.2 |
| KKT4 | 1556 | 33 | 46.8 |
| KKT14 | 1098 | 29 | 31.5 |
| KKT10 | 908 | 40 | 37.2 |
| KKT19 | 724 | 34 | 36.0 |
| KKT11 | 688 | 24 | 49.0 |
| **Tb927.8.2830 (KKT25)** | 684 | 32 | 42.2 |
| KKT6 | 565 | 20 | 45.4 |
| KKT5 | 481 | 19 | 34.2 |
| KKIP5 | 377 | 7 | 16.9 |
| KKT3 | 302 | 12 | 17.6 |

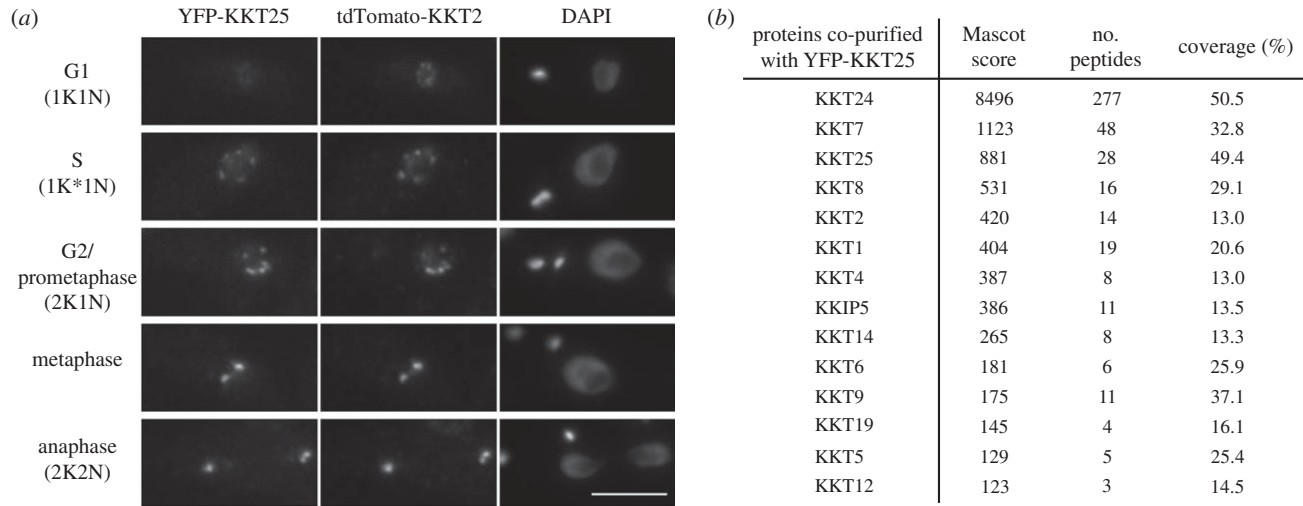

**Figure 3.** Identification of KKT25. (*a*) YFP-KKT25 localizes at kinetochores from S phase to anaphase. Fluorescent protein signals were directly detected by microscopy. Bar, 5 μm. Cell line, BAP1820. (*b*) KKT25 co-purifies with a number of KKT proteins and KKIP5. BAP1742.

Figure 3(b) table:

| proteins co-purified with YFP-KKT25 | Mascot score | no. peptides | coverage (%) |
|---|---|---|---|
| KKT24 | 8496 | 277 | 50.5 |
| KKT7 | 1123 | 48 | 32.8 |
| KKT25 | 881 | 28 | 49.4 |
| KKT8 | 531 | 16 | 29.1 |
| KKT2 | 420 | 14 | 13.0 |
| KKT1 | 404 | 19 | 20.6 |
| KKT4 | 387 | 8 | 13.0 |
| KKIP5 | 386 | 11 | 13.5 |
| KKT14 | 265 | 8 | 13.3 |
| KKT6 | 181 | 6 | 25.9 |
| KKT9 | 175 | 11 | 37.1 |
| KKT19 | 145 | 4 | 16.1 |
| KKT5 | 129 | 5 | 25.4 |
| KKT12 | 123 | 3 | 14.5 |

trypanosome cells led to co-purification of many nuclear proteins including KKT proteins and KKIP2–7 [14]. However, KKT proteins were not detected in the KKIP1 immunoprecipitate without cross-linking [15], so the relationship between KKIP1 and KKT proteins remained unclear. While re-searching our previous mass spectrometry data [12] against a newer version of the *T. brucei* proteome database, we found that KKIP1 was actually present in the immunoprecipitate of KKT2 (figure 4*a*). We had therefore named it KKT21 but switched to use the name KKIP1 following the publication of the D'Archivio & Wickstead paper. Immunoprecipitation of KKIP1 using our protocol revealed co-purification with a number of KKT proteins (figure 4*b*), showing that KKIP1 is a genuine kinetochore protein. It is important to mention that we did not detect KKIP2–7 in our KKIP1 immunoprecipitation sample. This raises a possibility that previous identification of KKIP2–7 in the immunoprecipitate of KKIP1 from cross-linked cells was due to the artificial chemical cross-linking, which is consistent with the identification of many nuclear proteins in the same sample [14].

## 2.5. KOK subunits co-purify with a number of proteins with RNA-related functions

Immunoprecipitation of KKIP2–7 from non-cross-linked cells identified a complex called the KOK complex that consists of nine KKIP proteins (KKIP2, 3, 4, 6, 8, 9, 10, 11 and 12) and localizes at the outer region of kinetochores during mitosis [15]. It has been proposed that KKIP1 provides a linkage between inner kinetochores and the KOK complex, despite the fact that KKIP1 was not detected in the immunoprecipitates of any KOK components [15]. To reveal the relationship between the KOK complex and KKT proteins, we performed immunoprecipitation of KKIP2–7 using our purification protocol. Immunoprecipitation of KKIP7 did not reveal any specific interacting proteins (electronic supplementary material, table S1), while that of KKIP5 was unsuccessful despite multiple attempts (electronic supplementary material, table S1; additional data not shown). By contrast, immunoprecipitation of KKIP2, KKIP3, KKIP4 and KKIP6 revealed a number of co-purifying proteins, including the KOK components

| (a) | proteins co-purified with YFP-KKT2 | Mascot score | no. peptides | coverage (%) |
|---|---|---|---|---|
| | KKT2 | 1773 | 55 | 46.0 |
| | KKT1 | 1023 | 31 | 23.5 |
| | KKT7 | 701 | 22 | 45.5 |
| | KKT8 | 538 | 17 | 59.6 |
| | KKT11 | 302 | 8 | 32.3 |
| | KKT10 | 295 | 13 | 27.7 |
| | KKT16 | 291 | 4 | 43.3 |
| | KKT9 | 287 | 9 | 32.2 |
| | KKT6 | 238 | 6 | 28.3 |
| | **KKIP1** | 219 | 6 | 16.2 |
| | KKT4 | 173 | 3 | 11.5 |
| | KKT14 | 156 | 3 | 10.4 |
| | KKT17 | 138 | 4 | 17.2 |
| | KKT19 | 129 | 8 | 22.7 |
| | KKT15 | 127 | 2 | 22.8 |

| (b) | proteins co-purified with YFP-KKIP1 | Mascot score | no. peptides | coverage (%) |
|---|---|---|---|---|
| | KKT1 | 4022 | 172 | 33.5 |
| | KKT7 | 3824 | 176 | 78.9 |
| | KKIP1 | 3607 | 182 | 61.3 |
| | KKT9 | 2765 | 98 | 75.1 |
| | KKT8 | 2216 | 81 | 80.7 |
| | KKT2 | 1626 | 79 | 48.1 |
| | KKT4 | 1529 | 58 | 54.9 |
| | KKT14 | 710 | 28 | 46.3 |
| | KKT6 | 641 | 24 | 42.4 |
| | KKT11 | 614 | 32 | 62.9 |
| | KKT10 | 448 | 25 | 33.5 |
| | KKT19 | 440 | 23 | 35.7 |
| | KKT5 | 404 | 23 | 40.4 |
| | KKT15 | 301 | 14 | 33.4 |
| | KKT3 | 203 | 18 | 17.9 |

**Figure 4.** KKIP1 co-purifies with KKT proteins, not with KKIP proteins. (a) Re-analysis of our previous mass spectrometry data [12] identifies KKIP1 in the KKT2 immunoprecipitate. (b) YFP-KKIP1 co-purifies with a number of KKT proteins but with none of other KKIP proteins. Cell line, BAP710.

(figure 5a–d). Besides Tb927.3.3740 and Tb927.2.3160/Gar1, which were detected in the previous report [15], 10 additional proteins were identified as apparent interactors of KKIP2/3/4/6, many of which have putative domains implicated in RNA binding, transcription or splicing (figure 5e). Interestingly, our HHpred analysis also revealed a similarity to a CTD kinase subunit in KKIP6 as well as a putative RNA recognition motif in KKIP4, KKIP9 and KKIP10 (figure 5e). The functional significance of these factors for kinetochore functions, if any, remains to be determined. Although we did not detect significant amounts of KOK components in our immunoprecipitates of KKT proteins or KKIP1, several kinetochore proteins were detected in the immunoprecipitates of KOK components, especially KKIP3 (figure 5b). Because KKIP1 did not co-purify with any KOK components using the same purification protocol (figure 4b), kinetochore localization of the KOK complex may be mediated by KKIP3, rather than KKIP1.

# 3. Discussion

In this study, we identified four additional kinetochore components in T. brucei (KKT22–25). We also confirmed that KKIP1 is a genuine kinetochore protein. It is possible that KKIP5 is also a kinetochore protein based on its presence in the immunoprecipitates of KKT24 and KKT25 (figures 2b and 3b) as well as observed chromosome segregation defects upon depletion of KKIP5 [27]. Our original definition of genuine kinetochore proteins in T. brucei was that any such protein should co-purify 'only' with other kinetochore proteins (except for KKT4 and KKT20 that also co-purify with APC/C subunits) [12,13]. According to this definition, components of the KOK complex are not genuine kinetochore proteins because they co-purify with a number of factors that are implicated in RNA binding or processing. However, it has been clearly shown that KOK components localize at outer kinetochore regions at least during metaphase [14,15]. More importantly, our immunoprecipitation of KKIP3 revealed co-purification with several KKT proteins, suggesting that the KOK complex indeed localizes at kinetochores. Defects in chromosome segregation have not been reported after knockdown of KOK components or its interaction partners [15]. We speculate that the KOK complex

might be involved in the segregation of small chromosomes, rather than large chromosomes, in T. brucei.

The identification of a kinetochore protein that has a GNAT domain reinforces the idea that kinetoplastid kinetochores are unconventional. It will be important to test whether the GNAT domain of KKT23 is important for kinetochore functions. Acetylation of unknown substrates (possibly histones or kinetochore proteins) at centromeres might mark the position of kinetochore assembly sites in kinetoplastids that lack CENP-A. It is noteworthy that the genome-wide tagging project in T. brucei has come to an end, and did not identify any additional kinetochore components [28,29]. It is possible that we now have a complete list of kinetochore components, which include KKT1–20, KKIP1 and KKT22–25 (figure 6). Characterization of their functions and structures is not only important for our better understanding of eukaryotic chromosome segregation machinery but also for the development of new drugs against kinetoplastid diseases.

# 4. Material and methods

## 4.1. Cells

All cell lines, plasmids and primers/synthetic DNA used in this study are listed in electronic supplementary material, tables S2, S3 and S4, respectively. All cell lines used in this study were derived from T. brucei SmOxP927 procyclic form cells (TREU 927/4 expressing T7 RNA polymerase and the tetracycline repressor to allow inducible expression) [32]. Cells were grown at 28°C in SDM-79 medium supplemented with 10% (v/v) heat-inactivated fetal calf serum [33]. The cell line carrying KKT3-YFP as the sole copy of KKT3 was made by deleting one allele of KKT3 by a fusion PCR method [34] using a neomycin gene cassette from pBA183, followed by tagging of the remaining allele with a C-terminal YFP using a PCR-based method with a blasticidin selection marker [35]. N-terminally YFP-tagged KKT22 was made by a PCR-based method using pPOTv7 (eYFP, blasticidin) [35]. Endogenous YFP tagging for KKT23–25 and KKIP1–7 was performed using the pEnT5-Y vector [36] with PCR products or synthesized DNA fragments using XbaI/BamHI sites. Endogenous tdTomato tagging of

royalsocietypublishing.org/journal/rsob Open Biol. 9: 190236

(a)

| proteins co-purified with YFP-KKIP2 | Mascot score | no. peptides | coverage (%) |
|---|---|---|---|
| KKIP3 | 5752 | 270 | 85.8 |
| KKIP11 | 3596 | 164 | 64.3 |
| KKIP2 | 2157 | 71 | 30.0 |
| KKIP8 | 1625 | 72 | 40.5 |
| KKIP9 | 460 | 14 | 37.2 |
| KKIP10 | 258 | 12 | 37.3 |
| KKIP4 | 254 | 12 | 38.0 |
| KKIP6 | 204 | 4 | 8.2 |
| KKIP12 | 132 | 4 | 42.8 |
| Tb927.10.11600 | 90 | 3 | 12.4 |
| Tb927.3.3740 | 76 | 2 | 10.0 |
| KKT1 | 71 | 5 | 7.8 |
| KKT7 | 62 | 3 | 12.3 |

(b)

| proteins co-purified with YFP-KKIP3 | Mascot score | no. peptides | coverage (%) |
|---|---|---|---|
| KKIP3 | 3357 | 157 | 78.5 |
| KKIP11 | 3222 | 142 | 66.5 |
| KKIP2 | 1571 | 44 | 26.8 |
| KKT1 | 880 | 27 | 23.3 |
| KKIP8 | 855 | 37 | 31.6 |
| KKT2 | 684 | 21 | 30.2 |
| KKT7 | 615 | 34 | 53.7 |
| KKT4 | 458 | 16 | 40.8 |
| KKIP9 | 456 | 16 | 34.5 |
| KKIP4 | 316 | 17 | 35.7 |
| KKIP10 | 312 | 12 | 32.4 |
| KKT14 | 257 | 10 | 20.0 |
| Tb927.10.11600 | 231 | 8 | 20.2 |
| KKIP6 | 164 | 4 | 15.1 |
| Tb927.3.3740 | 164 | 7 | 28.7 |
| KKT6 | 152 | 3 | 29.3 |
| Tb927.7.5590 | 136 | 4 | 11.6 |
| KKT24 | 118 | 5 | 13.2 |
| KKIP12 | 118 | 2 | 15.6 |

(c)

| proteins co-purified with YFP-KKIP4 | Mascot score | no. peptides | coverage (%) |
|---|---|---|---|
| KKIP4 | 9534 | 367 | 72.3 |
| KKIP9 | 8129 | 291 | 71.7 |
| KKIP10 | 6115 | 292 | 68.2 |
| Tb927.3.3740 | 5233 | 231 | 85.0 |
| Tb927.10.11600 | 4158 | 184 | 58.4 |
| KKIP12 | 4141 | 143 | 80.4 |
| Tb927.7.5590 | 3609 | 164 | 35.3 |
| KKIP6 | 3536 | 145 | 64.0 |
| Tb927.11.7450 | 2078 | 96 | 55.4 |
| Tb927.11.2900 | 1596 | 45 | 77.8 |
| Tb927.2.3160 | 757 | 29 | 43.6 |
| Tb927.8.7400 | 592 | 30 | 20.0 |
| Tb927.10.170 | 591 | 28 | 47.3 |
| Tb927.5.3250 | 577 | 21 | 18.1 |
| Tb927.7.6320 | 522 | 15 | 20.8 |
| KKIP11 | 489 | 20 | 26.1 |
| KKIP8 | 489 | 13 | 20.9 |
| KKIP3 | 408 | 22 | 40.9 |
| Tb927.10.4740 | 330 | 17 | 87.3 |

(d)

| proteins co-purified with YFP-KKIP6 | Mascot score | no. peptides | coverage (%) |
|---|---|---|---|
| KKIP9 | 6003 | 172 | 76.7 |
| KKIP4 | 5933 | 218 | 76.0 |
| KKIP10 | 5825 | 216 | 65.3 |
| KKIP12 | 3551 | 85 | 82.4 |
| Tb927.7.5590 | 2120 | 82 | 30.5 |
| Tb927.10.11600 | 1941 | 77 | 54.4 |
| Tb927.3.3740 | 1783 | 106 | 79.4 |
| KKIP6 | 1780 | 57 | 55.6 |
| Tb927.11.7450 | 588 | 26 | 30.1 |
| Tb927.5.3250 | 586 | 18 | 20.7 |
| Tb927.11.2900 | 584 | 18 | 60.8 |
| Tb927.4.5020 | 555 | 20 | 16.5 |
| Tb927.10.170 | 529 | 19 | 42.4 |
| KKIP3 | 497 | 23 | 40.7 |
| KKIP11 | 483 | 20 | 30.9 |
| Tb927.10.4740 | 310 | 14 | 87.3 |

(e)

| name | annotation and domains |
|---|---|
| Tb927.10.11600 | **putative splicing factor, arginine/serine-rich domain** |
| Tb927.3.3740 | **zinc-finger double-stranded RNA-binding** |
| Tb927.7.5590 | domain of unknown function (DUF3883) |
| Tb927.11.7450 | HIT zinc finger |
| Tb927.11.2900 | HIT zinc finger |
| Tb927.2.3160 | **H/ACA ribonucleoprotein complex subunit, Gar1/Naf1 RNA binding region** |
| Tb927.10.4740 | **H/ACA ribonucleoprotein complex subunit Nop10** |
| Tb927.10.170 | **H/ACA ribonucleoprotein complex subunit Cbf5, tRNA pseudouridine synthase** |
| Tb927.5.3250 | **weak similarity to pre-mRNA-splicing factor 8** |
| Tb927.7.6320 | BTB/POZ domain, regulator of chromosome condensation (RCC1) repeat |
| Tb927.8.7400 | **RNA polymerase IIA largest subunit** |
| Tb927.4.5020 | **DNA-directed RNA polymerase II subunit RPB1** |
| | |
| KKIP2 | hypothetical protein |
| KKIP3 | putative PDZ domain |
| KKIP4 | **putative RNA recognition motif (RRM)** |
| KKIP6 | **weak similarity to CTD kinase subunit gamma** |
| KKIP8 | **poly(A) polymerase** |
| KKIP9 | **putative RNA recognition motif (RRM)** |
| KKIP10 | **putative RNA recognition motif (RRM)** |
| KKIP11 | hypothetical protein |
| KKIP12 | **RBP34, RNA-binding protein, putative RNA recognition motif (RRM)** |

**Figure 5.** KOK subunits co-purify with a number of proteins with RNA-related functions. (a–d) Mass spectrometry summary tables of KKIP2, KKIP3, KKIP4 and KKIP6 immunoprecipitates. Cell lines, BAP825, BAP826, BAP808 and BAP828. (e) Putative domains in the identified proteins and KOK subunits.

KKT2 was performed using pBA164 that has a blasticidin selection marker [13] or pBA809 that has a neomycin marker. pBA809 was made by subcloning of the KKT2 targeting fragment from pBA67 [12] into pEnT6-tdTomato [36] using XbaI/BamHI sites. All constructs were sequence verified. Plasmids linearized by NotI were transfected to trypanosomes by

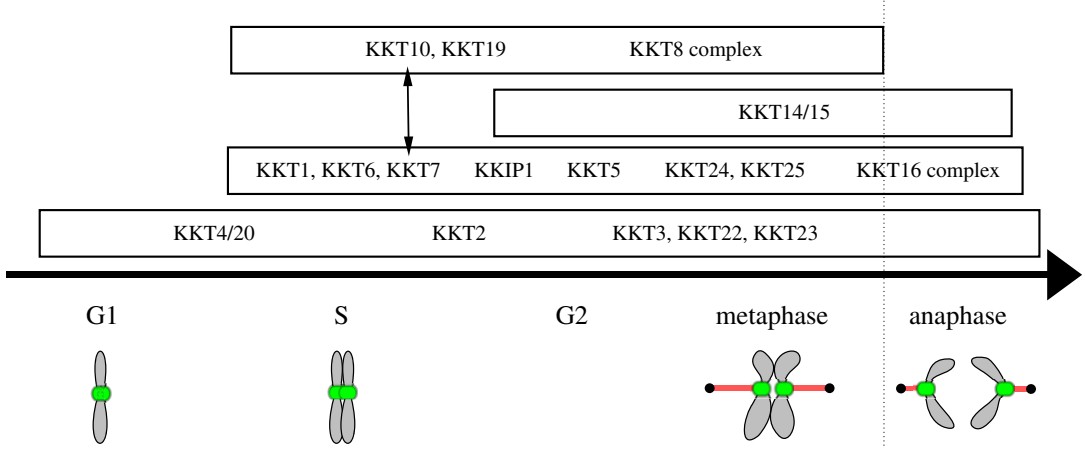

**Figure 6.** Localization patterns of kinetoplastid kinetochore proteins. Note that direct protein–protein interactions have not been established for many kinetochore proteins, except for the KKT7–KKT10 interaction (indicated by two-headed arrow) and the KKT8 complex that consists of KKT8, KKT9, KKT11 and KKT12 [30]. The putative KKT16 complex consists of KKT16, KKT17 and KKT18 [12]. S-phase-specific kinetochore protein KKT13 is not shown. Modified from [12,31] under the Creative Commons Attribution License.

electroporation into an endogenous locus. Transfected cells were selected by the addition of 25 µg ml⁻¹ hygromycin (pEnT5-Y derivatives), 10 µg ml⁻¹ blasticidin (pBA164 or pPOTv7-based PCR products) or 30 µg ml⁻¹ G418 (pBA809 or pBA183-based PCR products).

## 4.2. Fluorescence microscopy

Cells were fixed with 4% paraformaldehyde for 5 min and images were captured at room temperature on a DeltaVision fluorescence microscope (Applied Precision) installed with softWoRx v.5.5 housed in the Micron Oxford essentially as described [13] using 100× objective lenses (1.42 NA). Images were processed in ImageJ [37]. We confirmed that there was no notable bleed-through signal from different channels in our experiments. Typically, 12 optical slices spaced 0.25 µm apart were collected, and single plane images are shown. Figures were made using Inkscape (The Inkscape Team) and converted to the EPS format.

## 4.3. Immunoprecipitation and mass spectrometry

Immunoprecipitation was performed as previously described using mouse monoclonal anti-GFP antibodies (Roche, 11814460001) that had been pre-conjugated with Protein G magnetic beads (Thermo Fisher Scientific, 10004D) with dimethyl pimelimidate (Sigma, D8388) [12]. Mass spectrometry was also performed essentially as previously described using a Q Exactive (Thermo Scientific) at the Advanced Proteomics Facility, University of Oxford [12]. Peptides were identified by Mascot (Matrix Science) using a custom *T. brucei* proteome database that contains predicted proteins in TriTrypDB (v.4) [38] supplemented with predicted small proteins [39,40]. Proteins identified with at least two peptides were considered as significant and shown in electronic supplementary material, table S1.

## 4.4. Bioinformatics

Search for homologous proteins were done using BLAST in TriTrypDB [38,41], Jackhmmer on the UniProtKB proteome database using a default setting (HmmerWeb v.2.39 [42]) or hmmsearch on select proteomes using manually prepared hmm profiles (HMMER v.3.0 [43]). A protein with the best score in a given species was considered as a putative orthologue if a reciprocal search using BLAST or Jackhmmer identified the starting query protein as the best hit in the original species. HHpred was carried out using pfamA_v32.0 and PDB_mmCIF70 databases [20]. The multiple sequence alignment was performed with MAFFT (L-INS-i method, v.7) [44] and visualized with the Clustalx colouring scheme in Jalview (v.2.10) [45]. Coiled coils were predicted using COILS [46]. Accession numbers for protein sequences were retrieved from TriTrypDB [11,38,47–51] or UniProt [52].

Data accessibility. All data are available upon request.

Authors' contributions. O.O.N. performed immunoprecipitation of KKT3 and KKIP1–7, and identified KKT22 and KKT23. B.A. performed immunoprecipitation of KKT22, KKT23 and KKT25, and identified KKT24. P.L. performed immunoprecipitation of KKT24 and identified KKT25. All authors participated in data analysis. B.A. wrote the manuscript. All authors gave final approval for publication and agreed to be held accountable for the work performed therein.

Competing interests. We declare we have no competing interests.

Funding. P.L. was supported by Boehringer Ingelheim Fonds. B.A. was supported by a Wellcome Trust Senior Research Fellowship (grant no. 210622/Z/18/Z) and the European Molecular Biology Organization Young Investigator Program.

Acknowledgements. We thank Midori Ishii for comments on the manuscript and the Advanced Proteomics Facility, especially Svenja Hester and Sabrina Liberatori, for mass spectrometry analysis, as well as Micron Oxford. We also thank Samuel Dean for providing pPOT plasmids.

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

royalsocietypublishing.org/journal/rsob    *Open Biol.* **9**: 190236