## [Reviewer comments · Open Biology]

Review History

RSOB-19-0236.R0 (Original submission)

Review form: Reviewer 1

Recommendation

Accept with minor revision (please list in comments)

Do you have any ethical concerns with this paper?

No

Comments to the Author

Prior work from the Akiyoshi lab revealed the surprising finding that kinetoplastid species (such as Trypanosomes) lack the kinetochore components required for chromosome segregation in other eukaryotes. The Akiyoshi lab and complementary work in the Wickstead lab previously identified 20-30 proteins that act at Trypanosome kinetochores highlighting the unique features and evolution of this structure. This current paper now builds on that work to identify 4 additional Trypanosome kinetochore proteins using affinity purifications, mass spectrometry, and protein localization studies. This information is critical to provide a comprehensive understanding of this structure, its organization, and its unique function. Thus, the discovery of

these proteins represents an important contribution. The work described in this paper is technically solid and the identification of these proteins and the corresponding affinity purifications and localization are clear and compelling. Future work to analyze the functional contributions of these proteins will definitely be required to fully understand their relationships and roles in kinetochore function, but the existing extensive biochemical interaction studies provide an important advance that is worthy of publication in *Open Biology*. My one request is that the authors prepare a model figure for the end of the paper to highlight what they have learned on the subunit organization of the different *Trypanosoma* kinetochore complexes, their associations with each other, and a rough idea of overall kinetochore organization. There is a lot of great information contained here, but it is harder to interpret without a schematic diagram highlighting the different proteins and their relationships.

Review form: Reviewer 2

Recommendation

Accept with minor revision (please list in comments)

Do you have any ethical concerns with this paper?

No

Comments to the Author

While microtubule-dependent chromosome segregation is highly conserved in eukaryotes, and many kinetochore proteins are evolutionarily conserved from yeasts to animals, little has been known about the protein constitution of kinetoplastid species, including *Trypanosoma brucei*. Bungo Akiyoshi and colleagues have previously identified a panel of kinetochore proteins, KKT1-20, whose apparent homologs are not conserved beyond kinetoplastids. Identification and characterization of kinetoplastids are thus a highly interesting research subject, as it would shed light on an alternative design for a kinetochore architecture, and can help generate drugs that selectively target kinetoplastids without affecting other eukaryotes. This manuscript by the Akiyoshi lab reports identification of new four *T. brucei* kinetochore proteins, KKT22-25. This is supported by localization analysis and MS-based co-immunoprecipitation analysis. The authors further demonstrated that among KKIP proteins, a panel of kinetochore proteins previously identified by D'Archivio and Wickstead, only KKIP1 can coimmunoprecipitate with KKT proteins, while other KKIP proteins did not interact with KKTs.

Overall, the presented data support the major conclusions of the manuscript. Although immediate significance of the finding may be limited to the kinetoplastid field, this is unique characterization of kinetoplastid kinetochore proteins, forming a solid basis of future important study.

Minor points

1) The entire analysis is based on cell lines, where a tag is inserted to each endogenous gene of interest. The manuscript does not fully describe how the correct tagging is confirmed. Although MS-based identification of tagged protein is a piece of supporting evidence, it is concerning in the case of KKT22, for example, that YFP-KKT22 pull-down gave much lower recovery of KKT22, than YFP-KKT23 did. As this manuscript is essentially reporting the gene identification, I would expect to see genomic DNA analysis and immunoblotting analysis for correct tagging.

2) For the figures showing fluorescent images, I assume that fluorescent proteins were directly measured. This should be noted in figure legends. In methods, clarify if each image is a single

plane of deconvoluted image, or a projection of multiple planes. It would be also important to ensure that colocalization is not due to fluorescence bleed-through of a different channel.

3) Table S1 needs a more detailed explanation for data content.

Decision letter (RSOB-19-0236.R0)

18-Oct-2019

Dear Dr Akiyoshi

We are pleased to inform you that your manuscript RSOB-19-0236 entitled "Identification of four unconventional kinetoplastid kinetochore proteins KKT22-25 in *Trypanosoma brucei*" has been accepted by the Editor for publication in Open Biology. The reviewer(s) have recommended publication, but also suggest some minor revisions to your manuscript. Therefore, we invite you to respond to the reviewer(s)' comments and revise your manuscript.

Please submit the revised version of your manuscript within 7 days. If you do not think you will be able to meet this date please let us know immediately and we can extend this deadline for you.

- 1) A text file of the manuscript (doc, txt, rtf or tex), including the references, tables (including captions) and figure captions. Please remove any tracked changes from the text before submission. PDF files are not an accepted format for the "Main Document".
- 2) A separate electronic file of each figure (tiff, EPS or print-quality PDF preferred). The format should be produced directly from original creation package, or original software format. Please note that PowerPoint files are not accepted.
- 3) Electronic supplementary material: this should be contained in a separate file from the main text and meet our ESM criteria (see <http://royalsocietypublishing.org/instructions-authors#question5>). All supplementary materials accompanying an accepted article will be treated as in their final form. They will be published alongside the paper on the journal website

and posted on the online figshare repository. Files on figshare will be made available approximately one week before the accompanying article so that the supplementary material can be attributed a unique DOI.

Online supplementary material will also carry the title and description provided during submission, so please ensure these are accurate and informative. Note that the Royal Society will not edit or typeset supplementary material and it will be hosted as provided. Please ensure that the supplementary material includes the paper details (authors, title, journal name, article DOI). Your article DOI will be 10.1098/rsob.2016[last 4 digits of e.g. 10.1098/rsob.20160049].

4) A media summary: a short non-technical summary (up to 100 words) of the key findings/importance of your manuscript. Please try to write in simple English, avoid jargon, explain the importance of the topic, outline the main implications and describe why this topic is newsworthy.

Images

Data-Sharing

It is a condition of publication that data supporting your paper are made available. Data should be made available either in the electronic supplementary material or through an appropriate repository. Details of how to access data should be included in your paper. Please see <http://royalsocietypublishing.org/site/authors/policy.xhtml#question6> for more details.

Data accessibility section

Sincerely,

The Open Biology Team

<mailto:openbiology@royalsociety.org>

Reviewer(s)' Comments to Author:

Referee: 1

Comments to the Author(s)

Prior work from the Akiyoshi lab revealed the surprising finding that kinetoplastid species (such as Trypanosomes) lack the kinetochore components required for chromosome segregation in other eukaryotes. The Akiyoshi lab and complementary work in the Wickstead lab previously identified 20-30 proteins that act at Trypanosome kinetochores highlighting the unique features and evolution of this structure. This current paper now builds on that work to identify 4 additional Trypanosome kinetochore proteins using affinity purifications, mass spectrometry,

and protein localization studies. This information is critical to provide a comprehensive understanding of this structure, its organization, and its unique function. Thus, the discovery of these proteins represents an important contribution. The work described in this paper is technically solid and the identification of these proteins and the corresponding affinity purifications and localization are clear and compelling. Future work to analyze the functional contributions of these proteins will definitely be required to fully understand their relationships and roles in kinetochore function, but the existing extensive biochemical interaction studies provide an important advance that is worthy of publication in *Open Biology*. My one request is that the authors prepare a model figure for the end of the paper to highlight what they have learned on the subunit organization of the different *Trypanosoma* kinetochore complexes, their associations with each other, and a rough idea of overall kinetochore organization. There is a lot of great information contained here, but it is harder to interpret without a schematic diagram highlighting the different proteins and their relationships.

Referee: 2

Comments to the Author(s)

While microtubule-dependent chromosome segregation is highly conserved in eukaryotes, and many kinetochore proteins are evolutionarily conserved from yeasts to animals, little has been known about the protein constitution of kinetoplastid species, including *Trypanosoma brucei*. Bungo Akiyoshi and colleagues have previously identified a panel of kinetochore proteins, KKT1-20, whose apparent homologs are not conserved beyond kinetoplastids. Identification and characterization of kinetoplastids are thus a highly interesting research subject, as it would shed light on an alternative design for a kinetochore architecture, and can help generate drugs that selectively target kinetoplastids without affecting other eukaryotes. This manuscript by the Akiyoshi lab reports identification of new four *T. brucei* kinetochore proteins, KKT22-25. This is supported by localization analysis and MS-based co-immunoprecipitation analysis. The authors further demonstrated that among KKIP proteins, a panel of kinetochore proteins previously identified by D'Archivio and Wickstead, only KKIP1 can coimmunoprecipitate with KKT proteins, while other KKIP proteins did not interact with KKTs.

Overall, the presented data support the major conclusions of the manuscript. Although immediate significance of the finding may be limited to the kinetoplastid field, this is a unique characterization of kinetoplastid kinetochore proteins, forming a solid basis of future important study.

Minor points

1) The entire analysis is based on cell lines, where a tag is inserted to each endogenous gene of interest. The manuscript does not fully describe how the correct tagging is confirmed. Although MS-based identification of tagged protein is a piece of supporting evidence, it is concerning in the case of KKT22, for example, that YFP-KKT22 pull-down gave much lower recovery of KKT22, than YFP-KKT23 did. As this manuscript is essentially reporting the gene identification, I would expect to see genomic DNA analysis and immunoblotting analysis for correct tagging.

2) For the figures showing fluorescent images, I assume that fluorescent proteins were directly measured. This should be noted in figure legends. In methods, clarify if each image is a single plane of deconvoluted image, or a projection of multiple planes. It would be also important to ensure that colocalization is not due to fluorescence bleed-through of a different channel.

3) Table S1 needs a more detailed explanation for data content.

Author's Response to Decision Letter for (RSOB-19-0236.R0)

See Appendix A.

Decision letter (RSOB-19-0236.R1)

07-Nov-2019

Dear Dr Akiyoshi

We are pleased to inform you that your manuscript entitled "Identification of four unconventional kinetoplastid kinetochore proteins KKT22-25 in *Trypanosoma brucei*" has been accepted by the Editor for publication in Open Biology.

Article processing charge

Please note that the article processing charge is immediately payable. A separate email will be sent out shortly to confirm the charge due. The preferred payment method is by credit card; however, other payment options are available.

Sincerely,

The Open Biology Team

mailto: openbiology@royalsociety.org

Appendix A

We are very grateful to the Reviewers for their constructive comments on the manuscript and have addressed them as follows.

Referee: 1

My one request is that the authors prepare a model figure for the end of the paper to highlight what they have learned on the subunit organization of the different Trypanosome kinetochore complexes, their associations with each other, and a rough idea of overall kinetochore organization. There is a lot of great information contained here, but it is harder to interpret without a schematic diagram highlighting the different proteins and their relationships.

Response: We agree with the referee that a model figure would be helpful. At the same time, however, we feel that drawing a model about overall kinetochore organization at this point could be misleading because we still know little about the subunit organization or relative position of each kinetochore protein within the trypanosome kinetochore. Furthermore, our published data (e.g. the microtubule-binding kinetochore protein KKT4 may also bind DNA) and various unpublished observations suggest that the design principle of kinetoplastid kinetochores might be totally different from that of canonical kinetochores in other eukaryotes. Immunoprecipitation and mass spectrometry data are suggestive of some subcomplexes (e.g. KKT14/15 complex) but these have not been confirmed except for the KKT7-KKT10 interaction and the KKT8 complex that consists of KKT8/9/11/12 (Ishii and Akiyoshi, *bioRxiv* 2019). We would like to obtain more data before drawing a meaningful model of trypanosome kinetochores. Instead, we included Figure 6 that summarizes the localization pattern of kinetoplastid kinetochore proteins together with confirmed direct protein-protein interactions to help readers.

Referee: 2

Minor points

1) The entire analysis is based on cell lines, where a tag is inserted to each endogenous gene of interest. The manuscript does not fully describe how the correct tagging is confirmed. Although MS-based identification of tagged protein is a piece of supporting evidence, it is concerning in the case of KKT22, for example, that YFP-KKT22 pull-down gave much lower recovery of KKT22, than YFP-KKT23 did. As this manuscript is essentially reporting the gene identification, I would expect to see genomic DNA analysis and immunoblotting analysis for correct tagging.

Response: Thanks to the highly efficient homologous recombination in *Trypanosoma brucei*, we rarely encounter an issue of tagging wrong genes using a PCR-based method (with 80bp homology targeting regions) or a plasmid-based method (with 250bp homology targeting regions). We therefore do not routinely perform genomic DNA analysis any longer for YFP-tagging experiments. Regarding immunoblot analysis, it is often difficult to obtain robust signals for KKT proteins due to low protein levels. YFP-KKT22 described in the manuscript (strains BAP1454 and BAP1490) was made by a PCR-based method. We verified this result by making KKT22-YFP using a PCR-based method, as well as making YFP-KKT22 using a plasmid-based method (data not shown). Similarly, we tagged KKT23 by these three independent manners and got the same result. Together with mass spec data of YFP-KKT22–25 immunoprecipitates, we believe that right genes were tagged in this study. We included the following sentence into Figure 1 legend, “Similar results were obtained with KKT22-YFP and KKT23-YFP (data not shown)”.

2) For the figures showing fluorescent images, I assume that fluorescent proteins were directly measured. This should be noted in figure legends. In methods, clarify if each image is a single plane of deconvoluted image, or a projection of multiple planes. It would be also important to ensure that colocalization is not due to fluorescence bleed-through of a different channel.

Response: As requested, we added a sentence “Fluorescent protein signals were directly detected by microscopy.” to legends for Figures 1, 2, 3. In methods, we added the following sentence, “We confirmed that there was no noticeable bleed-through signal from different channels in our experiments. Typically, 12 optical slices spaced 0.25 μm apart were collected, and single plane images are shown.”

3) *Table S1 needs a more detailed explanation for data content.*

Response: We added a more detailed explanation for Table S1 in the supplementary material as follows. “Proteins identified with at least two peptides are shown. Common contaminants are marked by asterisk. Lack of KKIP2–7 peptides in the immunoprecipitate of KKIP1 and lack of KKIP1 peptides in the immunoprecipitates of KKIP2–7 are highlighted in green.”